# DistMLIP: A Distributed Inference Platform for Machine Learning Interatomic Potentials

**Kevin Han**
Carnegie Mellon University
kevinhan@cmu.edu

**Bowen Deng**[*]
UC Berkeley, LBNL
bowendeng@berkeley.edu

**Amir Barati Farimani**
Carnegie Mellon University
barati@cmu.edu

**Gerbrand Ceder**
UC Berkeley, LBNL
gceder@berkeley.edu

## Abstract

Large-scale atomistic simulations are essential to bridge computational materials and chemistry to realistic materials and drug discovery applications. In the past few years, rapid developments of machine learning interatomic potentials (MLIPs) have offered a solution to scale up quantum mechanical calculations. Parallelizing these interatomic potentials across multiple devices poses a challenging, but promising approach to further extending simulation scales to real-world applications. In this work, we present **DistMLIP**, an efficient distributed inference platform for MLIPs based on zero-redundancy, graph-level parallelization. In contrast to conventional spatial partitioning parallelization, DistMLIP enables efficient MLIP parallelization through graph partitioning, allowing multi-device inference on flexible MLIP model architectures like multi-layer graph neural networks. DistMLIP presents an easy-to-use, flexible, plug-in interface that enables distributed inference of pre-existing MLIPs. We demonstrate DistMLIP on four widely used and state-of-the-art MLIPs: CHGNet, MACE, TensorNet, and eSEN. We show that DistMLIP can simulate atomic systems 3.4x larger and up to 8x faster compared to previous multi-GPU methods. We show that existing foundation potentials can perform near-million-atom calculations at the scale of a few seconds on 8 GPUs with DistMLIP.

## 1 Introduction

Atomistic simulation has been the workhorse in computational materials and drug discovery over the recent years (Merchant et al., 2023; Jain et al., 2013; De Vivo et al., 2016). The chemical properties and behavior of a material are essentially determined by the interactions in the given set of atomic arrangements. In the most simplified framework, one can formulate this problem as solving the function that determines the potential energy surface (PES) of a set of atoms given by $E = \phi(\vec{r}_i, C_i)$, where $E$ is the energy, $\vec{r}_i$ and $C_i$ are the positions and chemical identities of the atoms.

To study the material's properties, multiple fundamentally different methods have been developed to obtain or construct the function $\phi$. Classical force fields (FF) like embedded atom methods (Daw & Baskes, 1984), CHARMM (Vanommeslaeghe et al., 2010), and Amber (Wang et al., 2004) qualitatively predict the PES and the bond energy between atoms. These classical FFs are cheap, intuitive, and explainable, but are often not accurate enough and have been constructed and applied to narrow chemical domains and a few elements. Fundamentally, the interactions in $\phi$ are determined by the electronic structure of a material and can be solved in first principles by quantum mechanics. Quantum chemical simulation methods, such as Density Functional Theory (DFT) (Perdew et al., 1996) and coupled cluster (CC) methods (Raghavachari et al., 1989), enabled the *ab initio* calculations of atomic behavior that are much more accurate than empirical methods. However, their computational complexity limits the practical use of quantum chemical simulation methods for many realistic applications. DFT, the most widely used *ab initio* simulation method, scales cubically $O(N_e^3)$ with

---

[*]Correspondence to Bowen Deng <bowendeng@berkeley.edu>

the number of electrons and is therefore limited to simulating only a few hundred atoms (Beck, 2000). Prohibitively high computational cost makes DFT only useful in describing materials properties that can be learned from a small simulation cell (Wang et al., 2024).

Machine learning approaches such as machine learning interatomic potentials (MLIPs) open the possibility to increase simulation scale while retaining quantum chemical accuracy by building ML surrogate models trained on DFT and CC data (Bartók et al., 2010; Zhang et al., 2018; Wang et al., 2024; Gasteiger et al., 2021; Deng et al., 2023; Batzner et al., 2022; Musaelian et al., 2023; Fu et al., 2025). Compared to feature-based classical FFs, deep-learning-based MLIPs enable improved learnability to better model the PES data. Graph neural networks (GNNs), especially, have demonstrated extraordinary computational efficiency and accuracy by learning both long-range and high-order atomic interactions through message passing. By design, the computation time of MLIPs scales linearly with the number of atoms $O(N)$, enabling simulations with tens of thousands of atoms at nano-second time scale.

Many materials engineering problems involve finite-sized effects like protein folding (Jumper et al., 2021), interfacial reactions (Du et al., 2023), particle-size effects (Shi et al., 2020), and formation of nano-domains (Holstun et al., 2025). Such systems require meso-scale simulations with upwards of millions of atoms, necessitating the ability to further scale the capacity of MLIP simulations. One promising solution is to expand MLIP inference from single-device to multi-device inference. Simulation packages such as LAMMPS provide *ad hoc* solutions for multi-GPU simulation (Thompson et al., 2022). Multi-device simulations are realized by dividing the total simulation cell into multiple, mutually exclusive, small cells for each device. Each small cell is then padded with additional atoms beyond the cell boundary in order to properly calculate the energy and forces within the small cell. This method, known as spatial partitioning, is based on the fundamental assumption that the force field only contains short-range interactions (Plimpton, 1995).

Since most MLIPs are designed to be relatively long-ranged, expanding the number of utilized GPUs during inference time is a nontrivial task due to the necessity to distribute the large system cell across multiple devices. Currently, there exists no native multi-GPU support for GNN-based MLIPs as most MLIPs have been implemented for only single-device inference. In order to support fast, accurate, and parallelizable atomistic simulations, we hereby present a distributed MLIP inference platform, **DistMLIP**, that enables efficient multi-device inference without the need for a modified architecture or additional training. Our highlighted contributions are as follows:

- DistMLIP features a simple, efficient, general, and versatile parallel inference platform for MLIP inference. By design, most popular MLIPs can be supported with a minimal amount of adaptation. In this work, we include benchmarking results of 4 widely used MLIPs: MACE, TensorNet, CHGNet, and eSEN.
- DistMLIP leverages **graph-level partitioning** that allows node and edge information to transfer between GPUs at each layer of the forward pass while still maintaining the intermediates required to perform backpropagation. This allows efficient parallelization of long-range GNN-based MLIPs, which is standard for most MLIPs today. Compared to spatial partitioning, graph partitioning has **zero redundancy**, meaning that no redundant computation is thrown away during parallel inference.
- We implemented the distribution of both the atom graph and the augmented three-body line graph, a common graph structure used in MLIPs to encode three-body atomic interactions.
- To allow flexible usage, DistMLIP does not depend on a 3rd party distributed simulation library such as LAMMPS. As a result, DistMLIP supports **plug-in usage** of any MLIP workflow.
- We show that the simple-yet-effective partitioning technique DistMLIP utilizes performs MD up to 8x faster compared with the more standard graph partitioning techniques.

## 2    RELATED WORK

### 2.1    MACHINE LEARNING INTERATOMIC POTENTIALS

The most common MLIP architecture today is GNN, where nodes in the atom graph represent atoms and edges in the atom graph represent the pair-wise distances between atoms that are within a

pre-defined cutoff distance (Batzner et al., 2022; Simeon & De Fabritiis, 2023; Passaro & Zitnick, 2023; Gasteiger et al., 2021; Schütt et al., 2021; 2018; Smith et al., 2017). GNN computation scales linearly with the number of atoms, as the amount of computation is associated with the neighbors within the receptive field of each atom. Some MLIPs also pass messages on top of higher-order graphs, such as threebody bond graphs, that encode angles as pairwise information between bonds (Choudhary & DeCost, 2021; Deng et al., 2023; Yang et al., 2024). MLIPs built on top of the transformer architecture have also been introduced, where "tokens" represent individual nodes and full self-attention is performed over all tokens (Liao et al., 2024; Vaswani et al., 2017). Recently, a class of foundation potentials (FPs) have been shown to generalize across diverse chemistries by pretraining on massive datasets (Chen & Ong, 2022; Deng et al., 2023; Chanussot* et al., 2021; Barroso-Luque et al., 2024; Yang et al., 2024; Merchant et al., 2023; Kaplan et al., 2025). These pretrained FPs substantially reduce the need for target-system training, and their open-sourced pretrained checkpoints serve as ready-to-use universal MLIPs.

## 2.2 Spatial Partitioning

**LAMMPS** implements multi-GPU inference via a spatial partitioning approach where the simulation space is split into mutually exclusive partitions. For each mutually exclusive partition, LAMMPS creates a second, larger partition that includes all atoms up to the interaction radius of the FF, commonly known as border or ghost nodes. This is required as the energy and force calculation of the atoms within each mutually exclusive partition requires the atomic information from all nodes within the model's interaction radius. This leads to highly redundant calculations as the computation performed on the ghost nodes is thrown away after each time step. By estimate, a 64-molecule water system calculated with a 6-layer GNN that has a 6 angstrom cutoff distance would require the computation of 20,834 ghost atoms when using spatial partitioning (Musaelian et al., 2023). Furthermore, unlike classical FFs, most MLIPs do not have a mature interface with LAMMPS, making spatial partitioning practically infeasible for the majority of MLIPs that have been developed.

**DeepMD** is a short-range MLIP that has been applied to the simulation of 100 million atoms of water by 27,360 NVIDIA V100 GPUs on the Summit supercomputer, utilizing the spatial partitioning features within LAMMPS (Jia et al., 2020). The atomic system size was further extended to 10 billion atoms after further optimization of model tabulation, kernel fusion, and redundancy removal of the **Deep Potential** architecture (Guo et al., 2022).

**Allegro** has been developed as a strictly local, E(3)-equivariant interatomic potential that features efficient parallelization through spatial partitioning due to its short-range design (Musaelian et al., 2023). Because of this strict locality, Allegro demonstrated good scaling on large atomic systems – Kozinsky et al. (2023) used Allegro and LAMMPS to simulate a bulk Ag model with 100 million atoms, achieving 0.003 microseconds/atom-timestep using 128 NVIDIA-A100-80GBs.

However, strictly local models experience key limitations. The need for efficient parallelization restricts their interaction range to only a few angstroms, preventing their use on systems that require the modeling of long-range interactions (Zhou et al., 2023; Song et al., 2024; Gong et al., 2025; Cheng, 2025; Anstine & Isayev, 2023). Furthermore, the short-range design prevents the MLIP's application from simultaneously learning diverse chemical environments. As the short-range MLIP's cutoff is often determined by the radial distribution function of one targeted material system, it is infeasible to determine a universal cutoff that efficiently works for many materials, which is becoming a common scenario with the increased interest in FPs. These problems raise the need for a simple, unified, and versatile API to parallelize MLIPs.

**SevenNet**, derived from the Nequip architecture (Batzner et al., 2022), is one of the first MLIPs that support graph-parallel inference (Park et al., 2024). A simulation of 112,000 atoms $Si_3N_4$ was demonstrated by distributing the 0.84 million parameter SevenNet-0 on 8 A100-80GB GPUs. However, its graph parallel algorithm is not easily transferable to other MLIP architectures and relies on the combination of TorchScript and LAMMPS, making it unapplicable to simulation tasks and workflows that are not built upon LAMMPS (Larsen et al., 2017; Ganose et al., 2025; Barroso-Luque et al., 2022; Ko et al., 2025).

In addition, graph-parallelization has been previously explored for training large GNN models Sriram et al. (2022). In comparison, the parallelization in MLIP inference poses a fundamentally different challenge compared to training. During training, the samples are almost always restricted to very

small-scale chemical systems due to the computational complexity of acquiring labels. In training, the goal of graph-parallelization is to increase MLIP model sizes and batch sizes. During inference, graph parallelization is applied for the simulation of a single large chemical system. As a result, the application of graph-parallelization in large-scale MLIP simulations remains an open challenge.

## 3 METHODS

Figure 1(a) denotes an overview of DistMLIP. Public MLIP models can be easily adapted to perform distributed large simulations with DistMLIP. The core infrastructure of DistMLIP is in the construction of graphs, subgraphs, and communication-related metadata.

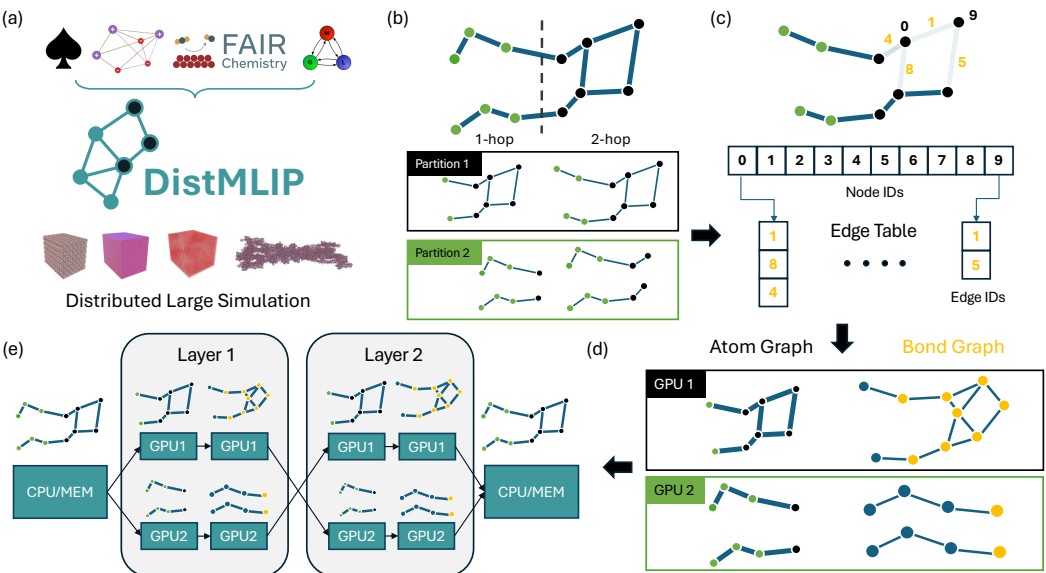

Figure 1: An overview of DistMLIP. **(a)** DistMLIP takes public MLIP models and performs large-scale, distributed simulations. **(b)** Partition the atom graph using a vertical spatial partitioning scheme, and construct subgraphs containing the 1-hop neighbors and 2-hop neighbors of the original partition, which are later used to calculate the distributed bond graphs. **(c)** Take the 2-hop atom graph and create an edge table backbone mapping node IDs (black) to edge IDs (orange) that contain the node ID as a source node. **(d)** Recursively traverse the edge table to construct the atom graph and bond graph. **(e)** Data transfer in a simple 2-layer graph neural network with both atom graph and bond graph.

### 3.1 GRAPH-PARALLEL MESSAGE PASSING

After the material system is converted into a graph using a neighbor list construction algorithm, the graph can be partitioned into subgraphs for each device, as illustrated in Fig. 1(b). At each graph convolution, each node's features are updated according to the edge and node features of its incoming neighbors. We can partition the nodes of the graph $G$ into $p$ disjoint sets, constructing graphs $G_1 \ldots G_n$, where $p$ is the number of partitions. Each of the graph partitions are distributed to its own GPU. To accurately calculate the features after one graph convolution, we expand $G_i$ into $G_i'$, where $G_i'$ consists of all nodes $v \in G$ such that there exists an edge $(v, u) \in E$ with $u \in G_i$. Formally,

$$G_i' = \{v \in V \mid \exists u \in G_i, (v, u) \in E\},$$

where $V$ and $E$ are the set of nodes and edges of the graph $G$, respectively. This ensures that all incoming information necessary for computing the convolution is included within $G_i'$. Let $H_i$ denote the set of all 1-hop nodes that were added to $G_i$ to create $G_i'$. Formally,

$$H_i = \{v \in V \setminus G_i \mid \exists u \in G_i, (v, u) \in E\}.$$

This represents the set of nodes in $V$ that are not in $G_i$ but have an outgoing edge into $G_i$. We refer to these nodes as border nodes. Let $E_i'$ denote the set of edges not in $G_i'$ that point to the border nodes.

---

**Algorithm 1** Atom Subgraph Creation

---

   **Input:** Atomic system nodes and edges
   **Output:** Partitioned subgraph with mappings
   **1. Create a partition rule based on the longest cell dimension (vertical walls)**
   **2. Assign atoms to buckets (PURE/TO/FROM)** using algorithm 3
   **3. Create node array and corresponding marker array for each partition:**
   **for** each starting partition $p_i$ (creating marker arrays) **do**
      initialize markers array
      markers$[0] = 0$
      markers$[1] = len(\text{PURE})$
      marker_index $= 0$
      **for** each destination partition $p_j$ **do**
         concatenate TO$[p_j]$ to $p_i$ node array
         markers[marker_index] = markers[marker_index - 1] $+ len(\text{TO}[p_j])$
         marker_index = marker_index $+ 1$
      **end for**
      **for** each source partition $p_k$ **do**
         concatenate FROM$[p_k]$ to $p_i$ node array
         markers[marker_index] = markers[marker_index - 1] $+ len(\text{FROM}[p_k])$
         marker_index = marker_index $+ 1$
      **end for**
   **end for**

---

Formally,
$$E_i' = \{(u,v) \in E \mid v \in H_i \text{ and } (u,v) \notin G_i'\}.$$
We refer to these edges as border edges, which we use extensively when distributing the bond graph.

After each graph convolution, we transfer the border node and border edge features to and from each partition, as shown in Fig. 1(d). After this transfer process, each GPU has the most updated node and edge feature to begin the next convolution. This implementation is completely model-agnostic and can be applied to both conservative and direct force prediction MLIPs.

### 3.2 DISTRIBUTING ATOM GRAPHS

In order to distribute the atom graph, we first partition the graph spatially using vertical wall partitions. Once these partitions are created, we specify algorithm 3 to identify the border nodes that each partition requires as well as the border nodes within each partition that other partitions require. For each partition, we create TO, FROM, and PURE arrays of node ids. We denote TO$_i[j]$ as the bucket of node ids associated with $G_i'$ required to be used in $G_j'$. Similarly, we denote FROM$_j[i]$ as the node ids associated with $G_j'$ required to be used in $G_i'$. As a result, TO$_i[j]$ and FROM$_j[i]$ should be the same array. The PURE bucket specifies the nodes that are not required in the data transfer process. Furthermore, each edge drawn from a border node to a pure node is assigned to the partition responsible for the pure node.

For each partition, we concatenate each of the arrays while maintaining a marker array containing the indices of the spans of each bucket. The marker array is used to efficiently index the spans of each of the features for data transfer between GPUs. The entire atom graph creation algorithm can be found in algorithm 1.

### 3.3 DISTRIBUTING HIGHER-ORDER GRAPHS

Higher-order graphs, sometimes referred to as line graphs or bond graphs (for the three-body case), are frequently used in MLIPs to featurize higher-order interactions Choudhary & DeCost (2021); Deng et al. (2023); Zhang et al. (2025). Distributing the bond graph involves selecting all 1-hop and 2-hop neighbors of the pure atom graph nodes assigned to a partition. We then create an edge table mapping from node ids to edges originating from the node id pointing to a different node. By recursively traversing the table, we are able to create the bond graph for each partition in parallel. Border nodes within the bond graph are associated with the 1-hop edge neighbors of border edges

within the atom graph – hence necessitating the inclusion of 2-hop neighbors. The parallel bond graphs thus contain the 1-hop neighbors of each pure bond graph node assigned to the partition. The complete procedure is found in algorithm 2 of the appendix.

DistMLIP graph creation runs purely on CPU memory, written in high-performance C. It is a standalone library that does not depend on external libraries such as LAMMPS, Pytorch, JAX, or Pytorch Geometric, and can be, in principle, applied to **any** MLIP that includes an atom graph and/or three-body graph.

### 3.4 CURRENTLY SUPPORTED MLIPs

Currently, we have implemented four widely used models in DistMLIP: 1) CHGNet (Deng et al., 2023), an invariant MLIP that features both an atom graph as well as a bond graph, 2) TensorNet (Simeon & De Fabritiis, 2023; Ko et al., 2025), an atom graph-only, computationally-efficient, equivariant MLIP with performance on par with Nequip, 3) MACE (Batatia et al., 2023), an atom graph-only equivariant MLIP that directly models many body interactions between atoms, and 4) eSEN (Fu et al., 2025), an atom graph-only, smooth and equivariant MLIP. Specific architecture details and usage are found in appendix C and appendix I.

## 4 RESULTS

In this section, we benchmark DistMLIP with the 4 FPs loaded with their public pretrained checkpoints: MACE-MP-0b-small-3.8M, CHGNet-2.7M, TensorNet-0.8M, and eSEN-3.2M. The details of the models and checkpoints can be found in appendix C.

For all scaling timing-related benchmarks, model inference is performed 20 times, with the average of the final 10 trials reported. This is to allow GPUs to warm up before performing calculations. We use crystalline $\alpha$-quartz $SiO_2$ supercells for each timing benchmark, unless stated otherwise. The benchmarks are performed with a GPU cluster with $8 \times$ NVIDIA-A100-80GB-PCIe.

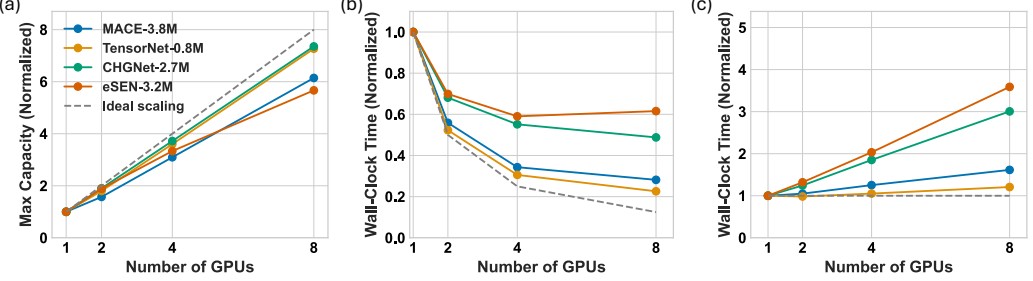

Figure 2: Performance scaling of DistMLIP inference with 4 pretrained MLIPs: MACE-3.8M, TensorNet-0.8M, CHGNet-2.7M, and eSEN-3.2M. All results are averaged over 10 inferences on a $SiO_2$ supercell. **(a)** Maximum capacity (number of simulatable atoms) vs. the number of GPUs. Values are normalized by the 1-GPU capacity. **(b)** Strong scaling of MLIP inference on DistMLIP, where the total number of atoms in the supercell is held constant while the number of GPUs increases. **(c)** Weak scaling behavior of MLIP inference on DistMLIP, where the number of atoms on each GPU device is held constant while the number of GPUs increases.

### 4.1 MAXIMUM CAPACITY

A key performance metric is the maximum number of atoms that can be simulated by extending to multi-GPU inference. The maximum capacity scaling tests, with respect to the number of GPUs, can be found in Figure 2(a). All atom counts are normalized to be represented as multiples of the 1-GPU maximum capacity. As the number of GPUs (and thus, total GPU memory) increases, the maximum simulatable capacity increases linearly. The scaling of eSEN and MACE is further away from ideal

scaling due to the one-time equivariant feature calculations that are occurring on a single GPU due to numerical stability concerns. The single GPU poses as a memory bottleneck for the system.

We also benchmark the maximum capacity and corresponding inference time against the SevenNet model. The results can be found in Appendix G. After matching the total number of parameters to SevenNet (800k), we find that MACE, TensorNet, and CHGNet can achieve up to 10x higher maximum capacity and 4x faster inference speed when incorporated with DistMLIP compared to the distributed inference of SevenNet.

## 4.2 Strong and Weak Scaling

Strong scaling tests, where the total size of the system remains constant while the number of GPUs increases, can be found in Figure 2(b). All times are normalized to be represented as multiples of the 1 GPU time. The system sizes for MACE-3.8M, TensorNet-0.8M, CHGNet-2.7M, and eSEN-3.2M were 33.5k, 22.0k, 9.8k, and 1.4k atoms respectively. We also plot the ideal scaling under the assumption that computation performed by each GPU is purely independent and perfectly parallelizable. In particular, eSEN's high memory consumption results in small atomic cells. However, small atomic cells with partition widths that aren't sufficiently large results in overlapping border nodes during each convolution – leading to increased overhead.

Weak scaling tests, where the total size of the system increases proportionally with the number of GPUs (such that each GPU performs computation on the same number of atoms), are found in Figure 2(c). All times are normalized to be represented as multiples of the 1 GPU time. The per GPU atom count for MACE-3.8M, TensorNet-0.8M, CHGNet-2.7M, and eSEN-3.2M are 34.6k, 19.9k, 9.9k, and 1.4k, respectively. In the case of eSEN-3.2M, weak scaling moves away from ideal scaling due to the high constant overhead associated with initial feature calculation. For CHGNet-2.7M, the computation required for the construction of the three-body graph scales with $O(N^6)$ where $N$ is the number of atoms within the three-body cutoff, leading to suboptimal weak scaling when the simulation cell size increases. In Table 3 of Appendix K.1, we show that the simple vertical partitioning rule used in DistMLIP and specified in Algorithm 3 is up to 8x faster compared to standard graph partitioning baselines.

## 4.3 Interaction Range

In this section, we benchmark how the parallelized simulation speed and capacity is affected by the MLIP's interaction range and number of parameters. In Fig. 3 (a), we fix each model to around 0.8M parameters and vary the number of message passing layers to increase the interaction range of the model. The 8 GPU inference is performed on the $\alpha$-quartz $SiO_2$ of 72k atoms. The measured inference times are then divided by the inference time of the baseline 10Å version of each MLIP. eSEN ran out of memory for the 45 and 50 angstrom tests.

The results in Fig. 3 (a) show that DistMLIP only has a linear relation between parallelized inference time vs. interaction range. This is due to the additional computation cost from each increased message passing layer. Conversely, in conventional spatial partitioning, the volume of the simulation cell, and therefore the number of ghost atoms, grows cubically with the interaction range. This highlights the parallelization efficiency and zero calculation redundancy in graph partitioning.

## 4.4 Scaling Model Size

In Fig. 3 (b) and (c), we fix the number of message passing layers and vary the feature embedding sizes in the MLIP, therefore measuring the relation between parallelized inference speed/capacity and model parameter size. The result shows that by decreasing the model parameter size, a significant increase in simulation speed and maximum capacity can be achieved. The result suggests an estimated performance gain when distributed inference can be combined with smaller model sizes through MLIP model distillation (Amin et al., 2025).

## 4.5 Real World Simulations

We also show the performance of real distributed simulations on a variety of solid-state and biomolecular systems, utilizing 1, 4, and 8 GPUs. The results are found in Table 1. We report

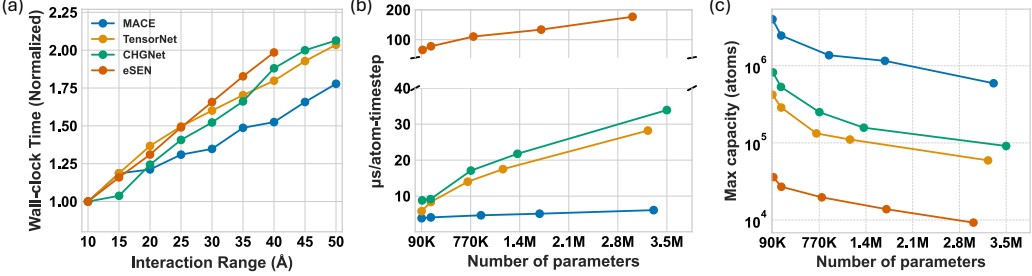

Figure 3: Effect of model configurations on graph-parallelized inference performance. **(a)** Inference time vs. MLIP interaction range while keeping model parameter size fixed. Values are represented as multiples of the 10Å interaction range. **(b)** Inference time and **(c)** maximum simulation capacity vs. number of parameters in the MLIP, while keeping interaction range fixed.

the microsceonds/atom-timestep of each model-system pair as well as the number of simulated atoms in the system. The simulated systems can be found in Figure 4. In Table 1, L-MACE-3.8M refers to multi-GPU inference of MACE using LAMMPS spatial partitioning, while the other 4 models are distributed with DistMLIP.

Table 1: MD step time (in $\mu s$ / (atom×step)) for the max capacity of 4 pretrained FPs on DistMLIP: MACE-MP-0b-small, TensorNet-MatPES-0.8M, CHGNet-MatPES-2.7M, eSEN-3.2M. L-MACE-3.8M refers to MACE running on LAMMPS spatial partitioning. L-MACE-3.8M is a compiled model using custom equivariant CUDA kernels while MACE-3.8M uses the pure-PyTorch implementation of MACE.

| Model | # GPUs | $\mu s$ / (atom×step) \| # of atoms (in thousands) | | | | |
|---|---|---|---|---|---|---|
| | | $Li_3PO_4$ | $H_2O$ | GaN | MOF | 2w49 |
| L-MACE-3.8M | 1 GPU | 82.47 \| 5.2 | 33.4 \| 10.4 | 19.8 \| 9.7 | 53.8 \| 8.0 | OOM |
| | 4 GPUs | 16.9 \| 41.4 | 10.1 \| 24.6 | 5.1 \| 45.0 | 9.4 \| 27.0 | OOM |
| | 8 GPUs | 12.3 \| 65.9 | 8.5 \| 82.9 | 2.7 \| 77.8 | 6.2 \| 64.0 | OOM |
| MACE-3.8M | 1 GPU | 44.8 \| 21.9 | 45.9 \| 20.7 | 39.5 \| 43.9 | 41.0 \| 16.0 | OOM |
| | 4 GPUs | 15.3 \| 110.6 | 18.2 \| 96.0 | 14.6 \| 128.0 | 14.7 \| 128.0 | 20.1 \| 69.3 |
| | 8 GPUs | 11.0 \| 216.0 | 11.6 \| 210.1 | 9.6 \| 250.0 | 10.9 \| 216.0 | 14.0 \| 69.3 |
| TensorNet-0.8M | 1 GPU | 81.7 \| 21.9 | 92.1 \| 6.1 | 79.1 \| 16.0 | 79.1 \| 16.0 | OOM |
| | 4 GPUs | 24.3 \| 64 | 26.9 \| 49.1 | 22.9 \| 65.5 | 23.2 \| 54.0 | OOM |
| | 8 GPUs | 16.3 \| 140.0 | 18.0 \| 82.9 | 15.9 \| 123.0 | 15.5 \| 125.0 | 19.6 \| 69.3 |
| CHGNet-2.7M | 1 GPU | 179.7 \| 4.1 | 154.8 \| 6.1 | 100.0 \| 5.5 | 174.6 \| 2.0 | OOM |
| | 4 GPUs | 94.8 \| 21.9 | 80.5 \| 20.7 | 45.5 \| 43.9 | 81.1 \| 16.0 | OOM |
| | 8 GPUs | 75.4 \| 46.7 | 64.5 \| 49.1 | 41.9 \| 77.8 | 67.1 \| 54.0 | OOM |
| eSEN-3.2M | 1 GPU | 727.3 \| 0.9 | 663.2 \| 1.3 | 438.9 \| 1.0 | 454.3 \| 1.0 | OOM |
| | 4 GPUs | 273.4 \| 4.1 | 284.0 \| 2.6 | 222.3 \| 5.5 | 236.3 \| 3.0 | OOM |
| | 8 GPUs | 241.2 \| 8.0 | 249.1 \| 6.1 | 198.9 \| 8.2 | 210.0 \| 6.0 | OOM |

Our result shows that DistMLIP provides tripled maximum simulation sizes compared to LAMMPS spatial partitioning within the MACE-3.8M model. Note that L-MACE-3.8M uses a compiled model with custom equivariant CUDA kernels, while DistMLIP MACE-3.8M only runs the pure-PyTorch implementation. Custom equivariant CUDA kernels were shown to accelerate MACE inference time

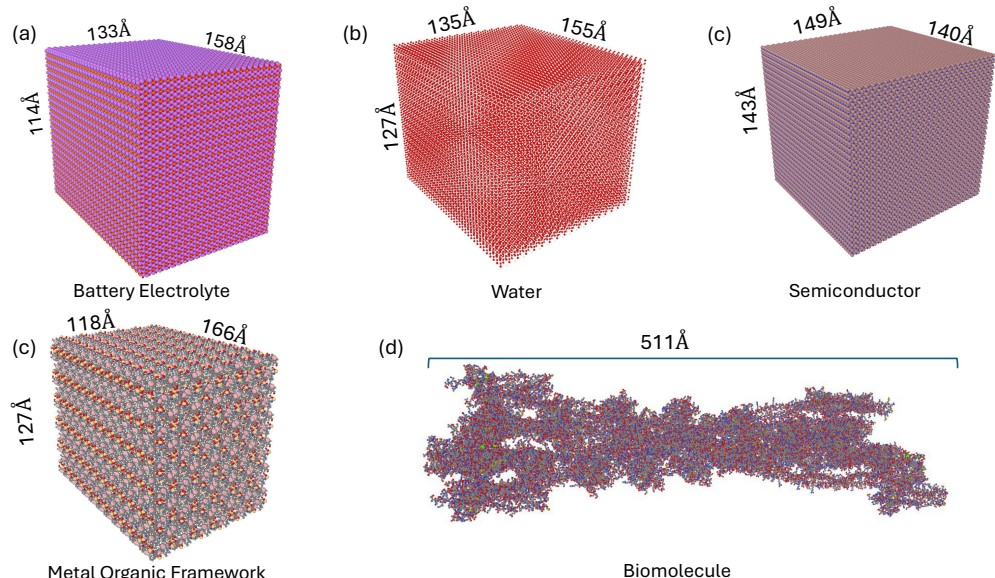

Figure 4: Sample simulation cells from real-world systems that are benchmarked in Table 1. **(a)** $Li_3PO_4$ supercell of 216.0k atoms. **(b)** $H_2O$ supercell of 210.1k atoms. **(c)** GaN supercell of 250.0k atoms. **(d)** $Cd_2B_2H_{48}C_{55}N_6(O_2F)_4$ metal organic framework (MOF) system of 216.0k atoms. **(e)** 2w49, an insect flight muscle protein of 69.3k atoms.

by up to 7.2x on large models (Geiger et al., 2024). Nevertheless, we observed similar simulation speed between the standard model on DistMLIP and the compiled model on LAMMPS, which supports the efficiency of DistMLIP graph-partitioning. No other model beyond MACE is reported due to the lack of LAMMPS multi-GPU inference support. For other FPs, which typically have longer interaction ranges compared to 12Å in MACE-3.8M, the capacity increase and inference speed-up should be much more significant as the efficiency of spatial partitioning degrades rapidly with increased cutoffs.

In Table 1, we highlight that most FPs with a few million parameters are capable of simulating near-million-atom scale systems when parallelized with only 8 GPUs. Moreover, we noticed that the inference time, when normalized by the number of atoms, is significantly decreased when any MLIP is being parallelized. This observation suggests chemically rare events can be cheaply simulated using a larger cell for a shorter simulation time, rather than a smaller cell for a longer simulation time, which has been the standard simulation procedure due to the inability to efficiently perform large simulations. As estimated from the benchmark result in Table 1, nanosecond near-million-atoms simulations can now be achieved at the order of 10 days with standard FPs and DistMLIP on a few GPUs.

## 5 CONCLUSION

Scaling quantum-chemical simulations to the size of realistic applications remains a critical challenge, even with recent developments of MLIPs and FPs. To address this challenge, we present DistMLIP, a distributed MLIP inference platform based on efficient graph-level partitioning. Compared to the conventional spatial partitioning through LAMMPS, DistMLIP serves as an easy and versatile distributed inference platform that supports long-range MLIPs. DistMLIP provides infrastructures for constructing and distributing atom and bond graphs, allowing the distribution of GNN-based MLIPs that are otherwise infeasible to parallelize.

We benchmarked the parallelized inference of 4 popular MLIPs: MACE, TensorNet, CHGNet and eSEN. Our result shows that efficient and plug-and-play parallelization can be achieved when combining DistMLIP with existing interatomic potentials. By distributing the MLIP simulation on 8 NVIDIA-A100 GPUs, our result shows that nanosecond, near-million-atom scale simulations can

be accomplished at the scale of 10 physical days with state-of-the-art FPs. We believe this effort to enable large-scale simulation would accelerate chemical, materials, and biological discovery.

## ACKNOWLEDGEMENTS AND DISCLOSURE OF FUNDING

This work was funded by the U.S. Department of Energy, Office of Science, Office of Basic Energy Sciences, Materials Sciences and Engineering Division under Contract No. DE-AC0205CH11231 (Materials Project program KC23MP). The authors would also like to thank Luis Barroso-Luque and Zijie Li for helpful discussions. The authors declare no competing interests.

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

## A   DISTRIBUTING BOND GRAPHS

Algorithm 2 depicts the method to distribute three-body graphs (bond graphs), as well as calculating the necessary information to perform data transfer between various partitions at each convolution of the three-body graph.

---

**Algorithm 2** Distributed Bond Graph Construction

---

**Input:** Global edges $E$, partitions $\{P_i\}$, bond cutoff $r$, tolerance $\tau$
**Output:** Line graphs $\{L_i\}$ for partitions $\{P_i\}$
**for** each partition $P_i$ **do**
   Initialize TO/FROM/PURE arrays for bond graph nodes (edges within atom graph)
   Initialize edge tables $T_i$ for each partition
   **Build Edge Table** $T_i$:
   **for** each edge $e \in E$ with dist$(e) \leq r + \tau$ **do**
     **if** dst$(e)$ in $P_i$ **then**
       append $e$ to $T_i[e.src]$
       **if** $e$ is border edge for $P_i$ **then**
         add $e$ to FROM$_{p_i}$[which_partition(e.src)]
       **else if** $e$ is border edge for another partition $P_j$ **then**
         add $e$ to TO$_{\text{which\_partition(e.src)}}[P_i]$
       **end if**
     **end if**
   **end for**
   **for** each edge $e \in E$ with dist$(e) \leq r + \tau$ **do**
     **if** $e$ is pure edge assigned to $P_i$ **then**
       Append $e$ to $T_i[e.src]$
       add $e$ to PURE[which_partition(e.dst)]
     **end if**
   **end for**
   **Localize Edges**
   **for** each $e \in T_i$ **do**
     Create mappings between global and local bond graph node indices
     Assign local node indices to each $e$ in $T_i$ $\forall i$
   **end for**
   **Build Line Graph** $L_i$
   **for** each partition $P_i$ **do**
     **for** each $v \in T_i$ **do**
       **for** each $e \in T_i[v]$ **do**
         **for** each $e' \in T_i[e.dst]$ **do**
           **if** needs_in_line$(e')$ **then**
             Draw an edge in bond graph from $e$ to $e'$ using local node indices
           **end if**
         **end for**
       **end for**
     **end for**
   **end for**
**end for**

---

## B  ASSIGN TO PARTITIONS

Algorithm 3 is the method used to determine assign individual nodes to the PURE/TO/FROM buckets of each partition. It is used extensively in both atom graph creation (algorithm 1) and three-body graph creation (algorithm 2).

---

**Algorithm 3** assign_to_partitions Subroutine

---

**Input:** Nodes, edges, partitions
**Output:** PURE, TO, FROM arrays for each partition
**1. Initialize node tracking:**
Create table `node_to_partition[node_id]` $\leftarrow$ -1 $\forall$ nodes
**2. Populating** `node_to_partition`
**for** each edge e **do**
   `node_to_partition` [which_partition$(e.src)$] =
     which_partition$(e.dst)$
**end for**
**3. Assigning nodes to partition buckets**
**for** each node n **do**
  **if** `node_to_partition`[n] $= -1$ **then**
    add n to PURE array of which_partition(n)
  **else**
    add n to $TO_{which\_partition(n)}$[`node_to_partition`[n]]
    add n to $FROM_{node\_to\_partition[n]}$[which_partition(n)]
  **end if**
**end for**

---

## C  MLIP VERSIONS IN BENCHMARK

The table below shows the checkpoint versions of the MLIPs tested. The CHGNet model is taken from recent release of MatGL library Ko et al. (2025). The eSEN model in our benchmark is not taken from the public pretrained checkpoints of 30.2M parameters, which is too big for efficient parallelized simulation. Instead, we initialized a 3.2M eSEN in accordance with the eSEN-MPTrj-3.2M configuration found in Fu et al. (2025).

Table 2: Pretrained MLIPs Model Specifications

| Model | Version | ModelSize | InteractionRange | Reference |
|---|---|---|---|---|
| CHGNet | matgl-MatPES-PBE-2025.2.10 | 2.7M | 45Å | (Deng et al., 2023) |
| MACE | MACE-MP-0b-small | 3.8M | 12Å | (Batatia et al., 2023) |
| TensorNet | matgl-MatPES-PBE-v2025.1 | 0.8M | 10Å | (Ko et al., 2025) |
| eSEN | eSEN-MPTrj-3.2M | 3.2M | 12Å | (Fu et al., 2025) |

## D  SINGLE GPU BENCHMARKING DETAILS

Because DistMLIP parallelizes neighbor list construction as well as underlying threebody graph creation, utilizing only 2 DistMLIP partitions can already lead to faster total inference time and less total memory consumption compared to a baseline implementation without DistMLIP (this is especially the case with CHGNet). Therefore, to maintain a fair comparison, all single-GPU results reported in any benchmark utilize 2 DistMLIP partitions performing operations on the same GPU. Therefore, only 1 GPU is utilized, but the same fast graph creation algorithms and implementation are shared. For all benchmarking tasks, 128 threads were used for neighbor list construction and graph creation.

## E    INFERENCE TIME BREAKDOWN

Neighbor list construction could take a substantial amount of inference time when the simulated system is large. In order to address this issue, we parallelized neighbor list construction in DistMLIP through multi-threading, so that graph creation time is substantially decreased compared to the single-thread neighbor list construction in Pymatgen (Ong et al., 2013). Furthermore, we show the breakdown of overall inference time using DistMLIP when fixing the total atomic system size in Figure 5. We also include the breakdown of overall inference time when fixing the total number of atoms per GPU while scaling the total number of GPUs in Figure 6. The atomic system used was a crystalline SiO2 supercell expanded in a cubic fashion.

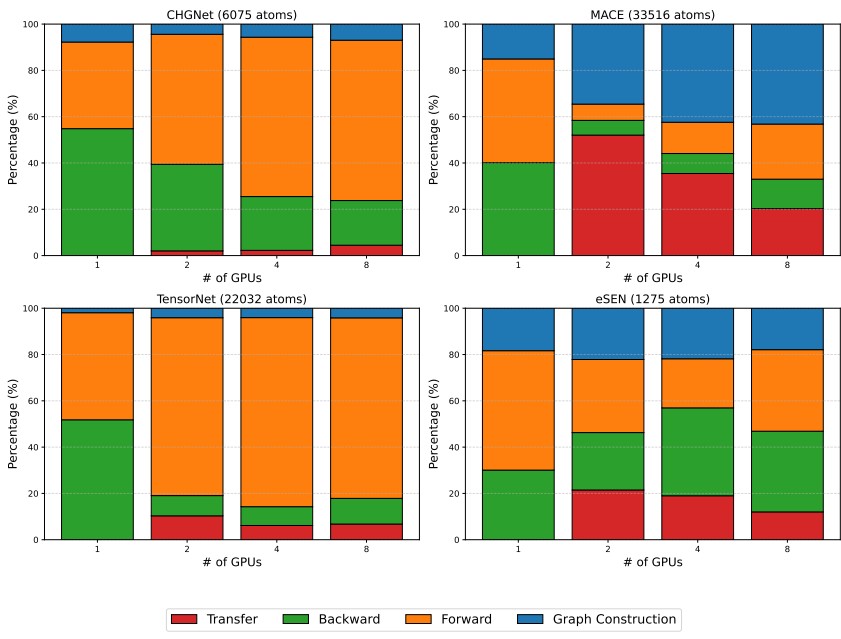

Figure 5: Timing breakdown, by percentage, for CHGNet-2.7M, MACE-3.8M, TensorNet-0.8M and eSEN-3.2M models across data transfer, backward pass (for force calculation), forward pass, and graph construction. The total number of atoms is held fixed across all GPUs runs.

## F    SCALING SYSTEM DENSITY

In Fig. 7, we plot the memory consumption and inference time of scaling system density (atoms/$\text{Å}^3$) of an $SiO_2$ system with 3456 atoms. DistMLIP inference with 4 A100-80GB GPUs were used. Denser atomic systems lead to a linear increase in total neighbor list size, driving up memory usage as well as inference time due to the decreased sparsity within the underlying atom graph's adjacency matrix. DistMLIP and its zero-redundancy inference algorithm scales memory consumption according to the increase in edge count.

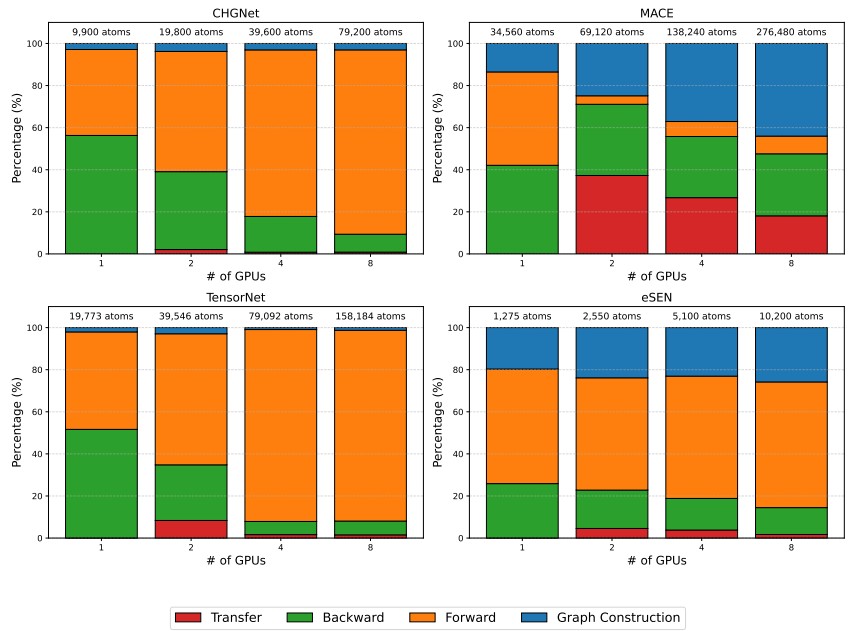

Figure 6: Timing breakdown, by percentage, for CHGNet-2.7M, MACE-3.8M, TensorNet-0.8M and eSEN-3.2M models across data transfer, backward pass (for force calculation), forward pass, and graph construction. The total number of atoms increase proportionally to the number of GPUs such that the number of atoms per GPUs is held fixed as the number of GPUs increases.

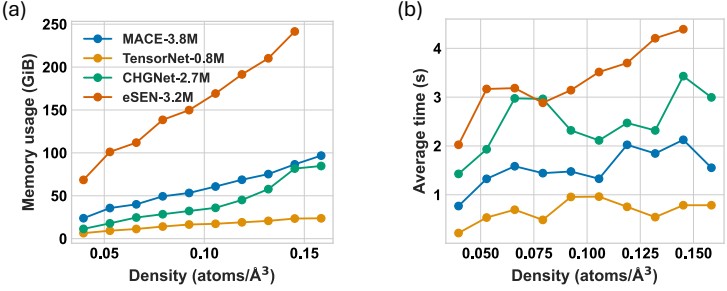

Figure 7: The effects of scaling density on **(a)** memory consumption, and **(b)** inference time. Both plots are the result of scaling atomic density (atoms/$\mathring{A}^3$) on an arbitrary system with fixed atom count using DistMLIP and 4 A100-80GB GPUs. eSEN is missing a datapoint due to out-of-memory issues.

## G  BENCHMARKING AGAINST SEVENNET

In this section, we benchmark the inference time and max capacity of the 4 MLIPs in DistMLIP against the distributed inference of SevenNet (Park et al., 2024). All the MLIPs are constructed to have a similar number of parameters as SevenNet-0 (0.8M parameters). All tests are performed on the supercells of the $\alpha$-quartz $SiO_2$. Inference times are averaged over 10 trials after 5 warmup trials.

Fig. 8 shows the result for (a)SevenNet, (b)MACE, (c)TensorNet, (d)CHGNet, and (e)eSEN. The number in each box in the heat map indicates the inference time of the given cell and the number of GPUs, and darker color represents faster inference. Grey boxes indicate the simulation failed due to the GPU out-of-memory error. We reproduced a similar maximum simulation size of 110k $\alpha$-quartz $SiO_2$ with SevenNet on 8 NVIDIA-A100-80GB, as indicated in the original manuscript. Our results indicated that MACE, TensorNet, and CHGNet can generally simulate larger maximum

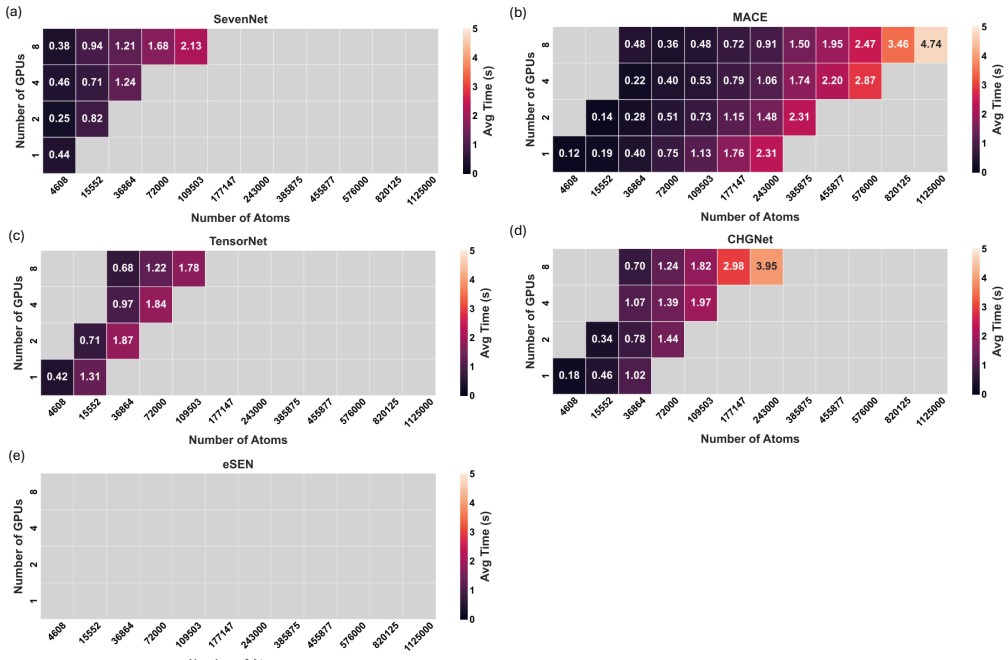

Figure 8: Inference speed and max capacity on $\alpha$-quartz $SiO_2$. Except for (a), all other models are distributed through DistMLIP. **(a)** SevenNet plus LAMMPS support. **(b)** MACE, **(c)** TensorNet, **(d)** CHGNet, and **(e)** eSEN. Because SevenNet is a 0.8M parameter model, all other DistMLIP models are initialized at 0.8M parameters for comparison purposes. Grey boxes denote the inability to simulate the system either due to out-of-memory issues or system-size issues.

capacity at faster speed in DistMLIP. For eSEN, all experiments failed due to the extensive memory consumption.

## H   ALIGNMENT OF SINGLE-DEVICE AND MULTI-DEVICE PREDICTIONS

DistMLIP's atom graph and bond graph distribution algorithms are exact in principle. However, numerical differences arise when performing computation in a distributed manner compared to on a single GPU. This is a result of non-determinism occurring during matrix multiplications and other operations on different GPUs (Fatahalian et al., 2004). Therefore, the exact same model and weights running single GPU inference on different GPUs within the same node will also yield slightly different results. In Fig. 9, we plot the energy/atom error in meV/atom units for MACE-3.8M, TensorNet-0.8M, and CHGNet-2.7M. The result shows that the numerical error from different GPUs is far below chemical accuracy.

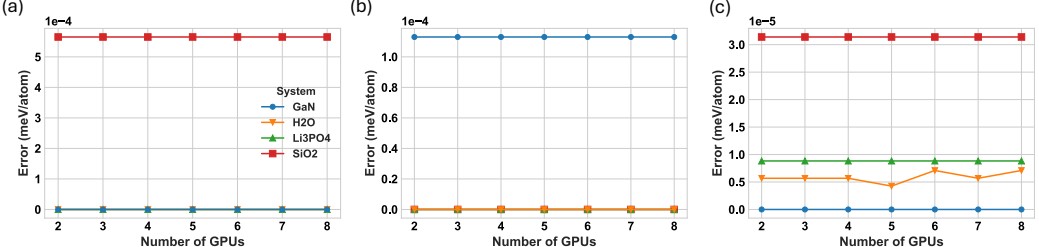

Figure 9: Energy (meV/atom) discrepancy between DistMLIP's multi-GPU inference and baseline single-GPU inference for **(a)** MACE-3.8M, **(b)** TensorNet-0.8M, and **(c)** CHGNet-2.7M on multiple chemical systems. Note that DistMLIP's graph partitioning and distribution algorithms are exact, and these non-perfect discrepancies are a result of non-deterministic matrix multiplication operations on different devices.

## I  USAGE

```
# Load the MLIP as usual
chgnet = ...
from DistMLIP.implementations.matgl import CHGNet_Dist
chgnet_dist = CHGNet_Dist.from_existing(chgnet)
chgnet_dist.enable_distributed_mode([0, 1, 2, ...]) # Specify GPU ids
# Run inference/simulation as usual
```

Code 1: Example code for using DistMLIP along with CHGNet. DistMLIP is designed to be a plug-and-play platform for distributed inference. The current implementation only supports single-node multi-GPU inference.

## J  PARALLELIZING A MODEL IN DISTMLIP

DistMLIP is designed to be both high-performant as well as easily usable. Parallelizing new MLIPs using DistMLIP is a straightforward process that can be done purely in Python. A few key points of DistMLIP are outlined in J.1. The primary data structure, the Distributed object, inputs atom positions, periodic boundary conditions, and (optionally) cell lattice, and constructs the graph partitions and associated metadata. The data structure can partition, aggregate, and perform data transfer for node features, edge features, and node features within the threebody graph. These simple distributed primitives are implemented in high performance C and makes distributing a new MLIP very straightforward to implement but still high performing.

### J.1  MODEL PARALLELIZATION EXAMPLE

```
# Creating a DistMLIP distributed object
dist = Distributed.create_distributed(...)
# Distributed edge information
dist.src_nodes, dist.dst_nodes # List of src and dist node pairs for each
    partition
# Distributing node/edge features
node_features_dist = dist.distribute_node_features(node_features)
edge_features_dist = dist.distribute_edge_features(edge_features)
# Exchanging node information
dist.atom_transfer(node_features_dist)
# Aggregating node features
node_features = dist.aggregate(node_features_dist)
```

Code 2: A subset of the available features implemented into DistMLIP. These features, implemented as a Python wrapper over efficient C and PyTorch code, allow for the straightforward distribution of any arbitrary MLIP.

# K   GRAPH PARTITIONING

Graph partitioning algorithms find applications in solving PDEs via domain decomposition, solving sparse linear systems of equations, circuit partitioning and layout, VLSI design, social network analysis, clustering algorithms, and image segmentation (Pothen, 1997; Stanton & Kliot, 2012; Bader et al., 2013; Tolliver & Miller, 2006; Peng et al., 2013). One common use case in graph partitioning is to create mutually exclusive spanning sets of nodes that contain the minimum number of edges that cross from one partition to another. This use case can be tackled using sparse, symmetric matrix reordering methods such as the Reverse Cuthill-Mckee (RCM) algorithm, which permutes a sparse matrix to minimize its bandwidth (i.e. reordering rows and columns such that non-zero values are closer to the diagonal) (Cuthill & McKee, 1969; Azad et al., 2017). The reordered matrix can then be partitioned along the columns in order to calculate graph node partitions. METIS is a graph partitioning algorithm that utilizes a graph coarsening phase, an initial partitioning sequence over the coarsened graph, and an uncoarsening and partition refinement stage (Karypis & Kumar, 1998). However, applications depending on algorithms such as RCM or METIS typically don't have latency requirements during the graph partitioning stage. In atomistic simulation, the underlying graphs are recalculated and repartitioned at each time step. Therefore, even small latency increases during the graph creation and partitioning stage get compounded into significant increases in overall simulation time. In our own experiments, we find that RCM and METIS could increase inference time for million-atom graphs by several seconds per timestep.

## K.1   BENCHMARKING PARTITION STRATEGIES

We compare DistMLIP's vertical wall partitioning strategy with other common graph partitioning algorithms. In Table 3, we replace DistMLIP's current vertical wall partitioning strategy with the Reverse Cuthill-McKee (RCMK) and METIS algorithms while holding the other components of Algorithm 1 and Algorithm 3 constant. Neither RCMK nor METIS supports threebody bond graph creation. RCMK and METIS both utilize the graph's topology in order to partition the graph such that the number of crossing edges between partitions is minimized. DistMLIP's current partitioning strategy, on the other hand, doesn't perform graph traversals but rather uses atomic positions as a heuristic in order to partition the graph. In Table 3, we also include the LAMMPS spatial partitioning results for comparison. As a result, we perform all benchmarks with the MACE-3.8M model.

Table 3: MD step time (in $\mu s$ / (atom$\times$step)) for various graph and spatial partitioning strategies. RCMK refers to the Reverse Cuthill-McKee algorithm used for graph partitioning. Both RCMK and METIS still utilize the DistMLIP platform, only the partitioning strategy is replaced. The model used was MACE-3.8M, and LAMMPS spatial partitioning values are included for comparison.

| Method | # GPUs | $\mu s$ / (atom$\times$step) \| # of atoms (in thousands) | | | | |
| --- | --- | --- | --- | --- | --- | --- |
| | | $Li_3PO_4$ | $H_2O$ | GaN | MOF | 2w49 |
| METIS | 4 GPUs | 81.19 \| 108.0 | 101.67 \| 96.0 | 68.98 \| 77.0 | 83.60 \| 125.0 | 78.93 \| 69.0 |
| | 8 GPUs | 62.18 \| 216.0 | 56.76 \| 216.0 | 77.15 \| 207.0 | 69.63 \| 216.0 | 67.10 \| 69.0 |
| RCMK | 4 GPUs | 77.49 \| 110.0 | 98.02 \| 96.0 | 66.92 \| 77.0 | 82.24 \| 125.0 | 79.94 \| 69.0 |
| | 8 GPUs | 57.22 \| 216.0 | 51.67 \| 216.0 | 74.01 \| 207.0 | 65.74 \| 216.0 | 65.51 \| 69.0 |
| Vert. wall | 4 GPUs | 15.30 \| 110.6 | 18.20 \| 96.0 | 14.60 \| 128.0 | 14.70 \| 128.0 | 20.10 \| 69.3 |
| | 8 GPUs | 11.00 \| 216.0 | 11.60 \| 210.1 | 9.60 \| 250.0 | 10.90 \| 216.0 | 14.00 \| 69.3 |

# L   MOLECULAR DYNAMICS SIMULATION STABILITY

To validate the numerical robustness and long-term stability of DistMLIP simulations, we performed a 2 nanosecond TensorNet MD simulation of a Li-ion cathode material containing 0.1 million atoms on 8 A100 GPUs. Fig. 10 shows the energy of the material as a function of time. Throughout the simulation, the system maintained structural stability. Except for the expected thermal fluctuation, no severe energy oscillation is observed during the entire simulation. The initial decrease in energy is not a numerical artifact; rather, it is attributed to the energy equilibration of the simulated material, representing a crucial physical process successfully captured by the DistMLIP simulation.

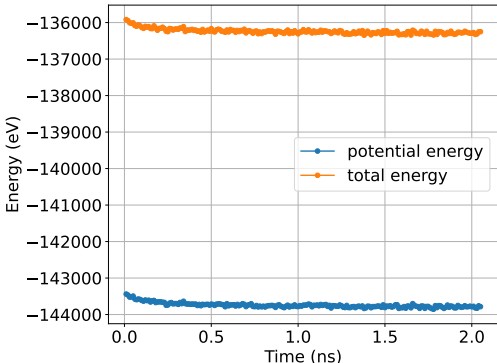

Figure 10: Evolution of potential energy and total energy in a 2 nanosecond long DistMLIP MD simulation of Li-ion cathode material using 8 GPUs and the TensorNet model. The long-time numerical stability of DistMLIP is indicated in the smooth profile of energies with only expected thermal fluctuations.

## M   LARGE SYSTEM TRAINING USING DISTMLIP

Machine learning interatomic potentials are trained on quantum mechanical calculations such as density functional theory or coupled cluster techniques. Due to the computational complexity of these techniques, however, calculating the energy and forces of atomic systems with greater than several hundred atoms is computationally intractable. As a result, other than instances in which large batch sizes are used in training, DistMLIP's primary use case would be for large scale inference. However, for technical completeness, we perform training on a GPCR protein, 6P9X from the protein data bank (Berman et al., 2000) consisting of 8.1k atoms. Energy and forces were calculated using a Lennard-Jones potential. With a batch size of 16, we achieve 2.1 seconds per training step on 8 A100-80 GB GPUs using a 0.4M parameter CHGNet model – demonstrating DistMLIP's ability in large-scale, large-batch training.

