# OpenReview forum: "DistMLIP: A Distributed Inference Platform for Machine Learning Interatomic Potentials"
_ICLR.cc/2026/Conference — ICLR 2026 Poster_

### Official Review · Reviewer_18ns · 2025-10-17

**Soundness:** 3
**Presentation:** 3
**Contribution:** 3
**Rating:** 8
**Confidence:** 4

**Summary:**

The authors propose a novel distributed inference platform, DistMLIP, for machine learning force fields.  DistMLIP enables efficient MLIP parallelization through graph partitioning, overcoming the requirement for LAMMPS. The results show that this platform 1)  significantly outperforms previous SOTA regarding the inference efficiency, 2) supports large-molecule inference, and 3) supports multiple network architectures.

**Strengths:**

1. DistMLIP significantly improves the inference speed for several models.
2. DistMLIP enables large molecular inference.
3. DistMLIP is model-agnostic.
4. DistMLIP achieves above features via full-atom level graph partition instead of spatial partition, omitting ghost atoms utilized by SevenNet, which are methodologically novel.

**Weaknesses:**

1. This platform does not support training.
2. This platform does not support multi-node inference.
3. The vertical wall is ad hoc. Although it performs well for the selected molecular system, it could fail for anisotropic molecules.

**Questions:**

Weakness 3 seems like an inherent limitation of DistMLIP by nature, but do you have any plan to address it, or to show that the method could also perform well for anisotropic molecules?

---

> ### Author Response · Authors · 2025-11-13
>
> Thank you for the positive review! We’re glad you were impressed with our results and inference platform. We will incorporate your feedback into the manuscript.
>
> >Weakness 3 seems like an inherent limitation of DistMLIP by nature, but do you have any plan to address it, or to show that the method could also perform well for anisotropic molecules?
>
> We find that vertical wall partitioning empirically performs quite well for MLIP inference for the molecular systems we’ve tried. Most importantly, vertical wall partitioning is simple and fast to perform. In Table 1, we show strong simulation performance on the 2w49 protein, a large anisotropic biomolecule consisting of 69.3k atoms – we’re able to simulate the protein using MACE with 4 GPUs while LAMMPS + MACE runs OOM even at 8 GPUs.

---

> > ### Comment · Reviewer_18ns · 2025-11-13
> > **Thank you for your reply**
> >
> > The response addresses my main concern, and I keep my score.

---

### Official Review · Reviewer_SsYg · 2025-10-29

**Soundness:** 2
**Presentation:** 2
**Contribution:** 2
**Rating:** 4
**Confidence:** 3

**Summary:**

DistMLIP is a distributed inference platform for ML interatomic potentials (MLIPs) that parallelizes graph-based models across GPUs with zero redundant compute. It partitions the atom graph (and the three-body “bond” line graph) and exchanges border features every GNN layer, so existing models (MACE, CHGNet, TensorNet, eSEN) can run multi-GPU without architectural changes. It can use up to 3.4× larger systems and is up to 8× faster than prior multi-GPU approaches.

**Strengths:**

1. Works with popular MLIPs (MACE, TensorNet, CHGNet, eSEN) with minimal adaptation, so we don’t need model-specific rewrites.
2. The “vertical” partition rule is reported up to 8x faster than standard graph partitioners (e.g., METIS/RCMK). And against SevenNet’s distributed inference, DistMLIP has up to 10x higher max capacity and is 4x faster.

**Weaknesses:**

Majors:
1. The authors say the design keeps backprop intermediates in their contribution claims, but they only benchmark inference; there’s no distributed-training result or accuracy/stability study over long MD runs.
2. All inference timing is on one cluster of 8x A100-80GB; there’s no multi-node or NVLink study to justify capability of large scale simulation.

Minors:
1. Line 39: "CHARM" -> "CHARMM"
2. Line 44: "coupled clustering" -> "coupled cluster"
3. Line 132: Citation format
4. Line 201: "G_n" -> "G_p"
5. Line 271: "Pytorch, Jax" -> "PyTorch, JAX"
6. Line 346: "partitoining" -> "partitioning"

**Questions:**

See weaknesses.

---

> ### Author Response · Authors · 2025-11-13
>
> Thank you for the review! We’re glad you see the strengths of our approach! We also want to thank you for finding typos within our manuscript – we will correct those errors.
>
> >The authors say the design keeps backprop intermediates in their contribution claims, but they only benchmark inference; there’s no distributed-training result or accuracy/stability study over long MD runs.
>
>
> MLIPs are trained on data generated using either DFT or CCSD quantum mechanical calculations. However, neither DFT nor CCSD methods are able to generate training data on the order of more than a few hundred atoms. As a result, MLIPs are almost never trained on large scale atomic system data because such data doesn’t exist. Therefore, the primary application of large scale MLIP calculation lies in inference. **We mention backprop intermediates in the contributions because all modern stable MLIPs require backprop in order to calculate forces [1-6]**. Forces are calculated as gradients of the energy prediction with respect to input positions. This means training requires second order methods.
>
> However, for technical completeness, we use DistMLIP to train a 0.4M CHGNet model using 8 A100-80GB GPUs on classical MD data of conformations of a GPCR protein consisting of 8.1k atoms and achieve 2.1 s/training step at batch size 16. We will include these results in the appendix of the manuscript.
>
> We do have a stability result that we will include in the appendix. We ran 8 A100-80GB GPUs with TensorNet for 2 nanoseconds (400k timesteps) on a 100k atom lithium ion cathode material and show no energy drift.
>
> [1] Fu, Xiang, et al. "Forces are not enough: Benchmark and critical evaluation for machine learning force fields with molecular simulations." arXiv preprint arXiv:2210.07237 (2022).
>
> [2] Fu, Xiang, et al. "Learning smooth and expressive interatomic potentials for physical property prediction." arXiv preprint arXiv:2502.12147 (2025).
>
> [3] Batatia, Ilyes, et al. "MACE: Higher order equivariant message passing neural networks for fast and accurate force fields." Advances in neural information processing systems 35 (2022): 11423-11436.
>
> [4] Batzner, Simon, et al. "E (3)-equivariant graph neural networks for data-efficient and accurate interatomic potentials." Nature communications 13.1 (2022): 2453.
>
> [5] Simeon, Guillem, and Gianni De Fabritiis. "Tensornet: Cartesian tensor representations for efficient learning of molecular potentials." Advances in Neural Information Processing Systems 36 (2023): 37334-37353.
>
> [6] Deng, Bowen, et al. "CHGNet as a pretrained universal neural network potential for charge-informed atomistic modelling." Nature Machine Intelligence 5.9 (2023): 1031-1041.
>
> >All inference timing is on one cluster of 8x A100-80GB; there’s no multi-node or NVLink study to justify capability of large scale simulation.
>
> DistMLIP was designed with running real-world, molecular dynamics use cases in mind. With 8 GPUs, we have shown that DistMLIP is already capable of efficiently simulating realistic atomic systems at significant scales similar to previously reported parallelized simulations that required hundreds to thousands of GPUs. We leave the additional algorithmic features to enable multi-node inference to be out of scope for the current work. With the current work, our goal is to first show that graph-parallel inference is substantially superior compared to previous distributed inference methods. Furthermore, most real-world practitioners do not have access to more than a single node of GPUs. Although we agree with the possible future need for mult-node simulations, we highlight our pioneering effort on this “sweet-spot” configuration that directly addresses the needs of the vast majority of simulation scenarios occurring in both academia and industry.

---

> > ### Comment · Reviewer_SsYg · 2025-11-13
> > **Official Comment by Reviewer SsYg**
> >
> > Thank you for your rebuttal; it resolves most of my concerns. Could you please share an updated version of the paper with all typographical errors corrected? I trust that you can complete the editing within this period.

---

> > > ### Author Response · Authors · 2025-11-14
> > >
> > > We're glad our rebuttal has resolved most of your concerns!
> > >
> > > We have just uploaded a revised version of the paper with all of the typographical errors corrected -- we would like to thank you for pointing them out!
> > >
> > > If you are satisfied with our rebuttal and revision of the paper, we would like to gently and kindly request for you to increase your score as you see fit. If you are still not satisfied, please do let us know so we can modify the manuscript at your request.

---

> > > > ### Comment · Reviewer_SsYg · 2025-11-15
> > > > **Official Comment by Reviewer SsYg**
> > > >
> > > > Thank you for your rebuttal and I have updated the score accordingly.

---

### Official Review · Reviewer_9NL9 · 2025-10-31

**Soundness:** 2
**Presentation:** 2
**Contribution:** 2
**Rating:** 4
**Confidence:** 3

**Summary:**

This paper introduces DistMLIP, a platform designed for efficient distributed inference of Machine Learning Interatomic Potentials (MLIPs), particularly targeting large-scale atomistic simulations (up to millions of atoms). The authors argue that while MLIPs offer quantum accuracy at lower cost, scaling them further requires multi-device parallelization . Existing methods, primarily based on spatial partitioning (like in LAMMPS), suffer from computational redundancy (ghost atoms) and are often ill-suited for modern, long-range GNN-based MLIPs . DistMLIP proposes a graph-level parallelization strategy based on graph partitioning, aiming for zero redundancy. It partitions the atom graph (and optionally higher-order graphs like bond graphs) across multiple GPUs and manages the communication of necessary node/edge features between partitions at each GNN layer . A key feature is its design as a flexible, easy-to-use, standalone platform that can wrap existing MLIPs without requiring model modification or reliance on specific simulation packages . The effectiveness is demonstrated on four diverse MLIPs (CHGNet, MACE, TensorNet, eSEN), showing significant improvements in maximum simulatable system size (up to 3.4x larger) and speed (up to 8x faster) compared to previous methods, enabling near-million-atom calculations in seconds on 8 GPUs.

**Strengths:**

- Scaling MLIP simulations to biologically and materially relevant sizes (millions of atoms) is a major challenge. DistMLIP provides a much-needed solution specifically tailored for efficient distributed inference of modern GNN-based MLIPs.
- DistMLIP is designed as a model-agnostic, plug-in platform . This allows researchers to apply it to their existing, pre-trained MLIPs with minimal adaptation (as demonstrated with four different models), significantly lowering the barrier to large-scale simulations. Its independence from specific simulation software like LAMMPS increases flexibility.
- The reported results are substantial: linear scaling of capacity, significant speedups in strong scaling tests (up to 8x faster), and the ability to perform near-million atom calculations in seconds on modest hardware (8 GPUs). Comparisons vs LAMMPS MACE and SevenNet further highlight the advantages.

**Weaknesses:**

- Graph partitioning inherently requires communication between GPUs after each message-passing layer to exchange border node information. The paper acknowledges scaling isn't always ideal (Fig 2b, 2c) partly due to overheads, but a more detailed analysis of communication cost vs. computation cost, and how it scales with the number of GPUs, graph density, and partition quality, would be valuable.
- The paper appears to be lacking comparisons against several relevant baselines. For instance, a critical benchmark is missing: how does the speed (e.g., throughput or latency) of the proposed system compare to other established distributed systems, such as Allegro?

**Questions:**

- Can the authors provide a breakdown of inference time into computation, communication, and graph construction/partitioning for different scenarios (varying GPU counts, system sizes)?
- What are the main challenges and potential strategies for extending DistMLIP to multi-node environments?
- How much effort is typically required to integrate a new MLIP model into the DistMLIP framework? Does it require modifications to the model's forward pass implementation? (Code 2 gives hints, but more context would be useful).

---

> ### Author Response · Authors · 2025-11-13
> **Response (part 1)**
>
> Thank you for the review! We’re glad to see you were impressed with the substantial improvements of DistMLIP over pre-existing systems such as LAMMPS and SevenNet, and we hope we can change your mind about the necessary contribution of our work. We will incorporate your feedback into the manuscript. Here are our responses to your weaknesses and questions sections:
>
> >Can the authors provide a breakdown of inference time into computation, communication, and graph construction/partitioning for different scenarios (varying GPU counts, system sizes)?
>
> Sure, here are 2 tables depicting precisely what you asked for:
>
> The breakdown of inference time as the number of GPUs increase while total number of atoms is held constant (strong scaling):
> | Model     |   Num Atoms |   Num GPUs |   Graph Construction |   Forward |   Backward |   Transfer |
> |:----------|------------:|-----------:|---------------------:|----------:|-----------:|-----------:|
> | chgnet    |        6075 |          1 |                 7.76 |     37.42 |      54.83 |       0    |
> | chgnet    |        6075 |          2 |                 4.41 |     56.15 |      37.41 |       2.04 |
> | chgnet    |        6075 |          4 |                 5.67 |     68.84 |      23.23 |       2.26 |
> | chgnet    |        6075 |          8 |                 6.97 |     69.26 |      19.29 |       4.48 |
> | tensornet |       22032 |          1 |                 1.94 |     46.25 |      51.8  |       0    |
> | tensornet |       22032 |          2 |                 4.1  |     76.84 |       8.74 |      10.32 |
> | tensornet |       22032 |          4 |                 4.06 |     81.68 |       8.12 |       6.15 |
> | tensornet |       22032 |          8 |                 4.16 |     77.98 |      11.09 |       6.78 |
> | mace      |       33516 |          1 |                15.06 |     44.79 |      40.15 |       0    |
> | mace      |       33516 |          2 |                34.57 |      7.02 |       6.37 |      52.04 |
> | mace      |       33516 |          4 |                42.4  |     13.51 |       8.63 |      35.45 |
> | mace      |       33516 |          8 |                43.19 |     23.81 |      12.71 |      20.3  |
> | esen      |        1275 |          1 |                18.35 |     51.58 |      30.08 |       0    |
> | esen      |        1275 |          2 |                22.15 |     31.57 |      24.79 |      21.49 |
> | esen      |        1275 |          4 |                21.84 |     21.2  |      37.97 |      18.99 |
> | esen      |        1275 |          8 |                17.91 |     35.22 |      34.86 |      12.02 |
>
> The breakdown of inference time as the number of GPUs and number of total atoms increase (weak scaling):
>
> | Model     |   Num Atoms |   Num GPUs |   Graph Construction |   Forward |   Backward |   Transfer |
> |:----------|------------:|-----------:|---------------------:|----------:|-----------:|-----------:|
> | mace      |       34560 |          1 |                13.53 |     44.37 |      42.11 |       0    |
> | mace      |       69120 |          2 |                24.86 |      4.04 |      33.83 |      37.26 |
> | mace      |      138240 |          4 |                37.09 |      7.12 |      29.06 |      26.73 |
> | mace      |      276480 |          8 |                44.02 |      8.41 |      29.48 |      18.09 |
> | tensornet |       19773 |          1 |                 2.05 |     46.3  |      51.64 |       0    |
> | tensornet |       39546 |          2 |                 2.93 |     62.28 |      26.43 |       8.36 |
> | tensornet |       79092 |          4 |                 0.92 |     91.18 |       6.26 |       1.63 |
> | tensornet |      158184 |          8 |                 1.28 |     90.64 |       6.56 |       1.52 |
> | chgnet    |        9900 |          1 |                 2.87 |     40.84 |      56.29 |       0    |
> | chgnet    |       19800 |          2 |                 3.79 |     57.13 |      37.01 |       2.07 |
> | chgnet    |       39600 |          4 |                 3.06 |     79.14 |      16.98 |       0.82 |
> | chgnet    |       79200 |          8 |                 3.05 |     87.58 |       8.5  |       0.86 |
> | esen      |        1275 |          1 |                19.68 |     54.44 |      25.87 |       0    |
> | esen      |        2550 |          2 |                23.86 |     53.34 |      18.24 |       4.56 |
> | esen      |        5100 |          4 |                23.06 |     58.12 |      15.01 |       3.82 |
> | esen      |       10200 |          8 |                25.82 |     59.72 |      12.77 |       1.7  |

---

> ### Author Response · Authors · 2025-11-13
> **Response (part 2)**
>
> >The paper appears to be lacking comparisons against several relevant baselines. For instance, a critical benchmark is missing: how does the speed (e.g., throughput or latency) of the proposed system compare to other established distributed systems, such as Allegro?
>
> Allegro is a model, rather than a distributed inference system, that is designed to have a small interaction radius to be more easily parallelizable. Allegro uses the LAMMPS framework in order to perform distributed simulation. We don’t believe comparing DistMLIP and Allegro is a logical comparison as it's unclear how to compare a distributed inference system with a model. Furthermore, we have already performed a comparison between LAMMPS and DistMLIP via the MACE model.
>
> >What are the main challenges and potential strategies for extending DistMLIP to multi-node environments?
>
> The primary challenge for extending DistMLIP to a multi-node environment involves efficient peer to peer communication between GPUs on different nodes. A multi-node implementation would require additional consideration on the specific network topology and how, physically, the nodes are interconnected. This additional consideration is necessary due to the stringent latency requirements for each individual timestep. We anticipate the new performance bottleneck to undoubtedly be network latency during inter-GPU feature transfers. However, we leave the additional algorithmic features to enable multi-node inference to be out of scope for the current work. With the current work, our goal is to first show that graph-parallel inference is substantially superior compared to previous distributed inference methods.
>
> >How much effort is typically required to integrate a new MLIP model into the DistMLIP framework? Does it require modifications to the model's forward pass implementation? (Code 2 gives hints, but more context would be useful).
>
> The effort to integrate a new MLIP model into DistMLIP is quite straightforward. The model’s forward pass implementation does need to be updated – but the update is almost as simple as adding dist.atom_transfer(node_feat) after each graph convolution. We thank you for the question and, as a result, will expand Code 2 and its corresponding section in the appendix to be more descriptive.
>
> Once again, we would like to thank the reviewer for the review!

---

### Official Review · Reviewer_dy5L · 2025-10-31

**Soundness:** 3
**Presentation:** 3
**Contribution:** 4
**Rating:** 8
**Confidence:** 3

**Summary:**

This paper introduces DistMLIP, a distributed inference platform designed to scale Machine Learning Interatomic Potentials (MLIPs) to large-scale atomistic simulations. The core problem it addresses is the computational bottleneck of running modern, high-accuracy MLIPs—many of which are based on Graph Neural Networks (GNNs) with long-range interactions—on systems with millions of atoms.

The key contribution is a "zero-redundancy, graph-level parallelization" strategy. This method contrasts with conventional spatial partitioning (e.g., in LAMMPS), which suffers from high computational redundancy due to the need to calculate "ghost atoms" at partition boundaries. DistMLIP partitions the atomic graph itself and distributes subgraphs to different GPUs, enabling efficient parallel inference. The platform is presented as a flexible, "plug-in" library that does not depend on third-party simulation packages like LAMMPS.

The authors demonstrate DistMLIP's effectiveness by benchmarking four popular MLIPs: CHGNet, MACE, TensorNet, and eSEN. The results show that DistMLIP can simulate systems 3.4x larger and achieve up to 8x faster performance compared to previous multi-GPU methods , enabling near-million-atom simulations on a single 8-GPU node.

**Strengths:**

### High-Impact Problem:

The paper tackles a critical and timely bottleneck in computational science. Scaling MLIPs to the meso-scale (millions of atoms) is essential for bridging quantum-accurate simulations with real-world applications in materials science, chemistry, and biology.

### Sound and Novel Method:

The graph-level parallelization approach is fundamentally better suited for GNN-based MLIPs than traditional spatial partitioning. The paper clearly articulates the "zero-redundancy" advantage , which correctly avoids the cubic scaling of redundant computations (ghost nodes) that spatial partitioning faces as the MLIP interaction range increases. The method's native support for both atom graphs and higher-order bond graphs (used in models like CHGNet) is a significant advantage.

- Comprehensive and Rigorous Empirical Validation: This is the paper's strongest aspect.

- Diverse Models: The method is validated on four distinct and widely-used MLIPs, demonstrating its generality.

- Strong Baselines: The authors provide direct comparisons against two crucial baselines: (1) The industry-standard spatial partitioning (LAMMPS-MACE) and (2) Another graph-parallel method (SevenNet). DistMLIP shows superior performance in maximum capacity and speed against both.

- Excellent Scaling Analysis: The paper provides clear strong and weak scaling plots (Fig. 2) , as well as detailed analyses of how performance scales with model parameters and, most importantly, interaction range (Fig. 3). The linear scaling with interaction range (vs. cubic for spatial partitioning) is a key result.

### Pragmatic Design Insight:

A standout finding is the justification for using a simple "vertical wall" partitioning scheme. The authors correctly identify that for MD, the partitioning latency (which must be paid at every time step) is a critical bottleneck . Their simple heuristic is shown to be up to 8x faster than more complex graph partitioning algorithms like METIS or RCMK (Table 4), demonstrating a deep, practical understanding of the problem domain.

### Practicality and Usability:

By designing DistMLIP as a standalone, "plug-and-play" Python-based library , the authors have significantly lowered the barrier to adoption for the broad community of researchers who use these models but are not experts in distributed computing.

**Weaknesses:**

### Single-Node Limitation:

 The paper states the current implementation only supports "single-node multi-GPU inference". This is a significant limitation for scaling to truly massive systems (tens of millions+ atoms), which would require a multi-node, multi-GPU setup. The paper would be stronger if it discussed the roadmap and key challenges (e.g., managing communication overhead of border node features across a network interconnect) for a multi-node implementation.

### Clarification of "8x Faster" Claim:

The abstract and introduction claim "up to 8x faster". However, the direct end-to-end inference comparison with SevenNet shows a ~4x speedup , and the LAMMPS-MACE comparison shows similar (not 8x faster) speeds, albeit with a non-compiled model. The 8x speedup figure appears to be sourced from the partitioning algorithm comparison in Table 4. The authors should clarify this in the main text to avoid overstating the end-to-end simulation speedup.

### Scaling of High-Order Graphs:

The paper honestly reports "suboptimal" weak scaling for CHGNet, attributing it to the three-body graph construction cost. This is an important detail, as it suggests that the performance benefits of DistMLIP may be partially bottlenecked by models with complex, high-order interactions. A brief discussion of whether this construction is (or can be) parallelized within DistMLIP would be beneficial.

### eSEN Performance:

The eSEN model consistently performs poorly, with high memory usage and frequent OOM errors. While this is likely due to the model's architecture rather than DistMLIP, its poor performance slightly detracts from the platform's "general and versatile" claim.

**Questions:**

## 1. Multi-Node Scalability:

The current work is an excellent demonstration of single-node, multi-GPU scaling. Could you elaborate on the primary challenges for extending DistMLIP to a multi-node environment? Specifically, how do you envision managing the atom_transfer step across a network interconnect, and what do you anticipate will be the new performance bottleneck (e.g., network latency vs. bandwidth)?

## 2. High-Order Graph Construction Bottleneck:

The suboptimal weak scaling of CHGNet is due to the three-body graph construction. Is this construction step (described in Algorithm 2) fully parallelized within DistMLIP, or is it a separate, serial (or partially parallel) step that acts as a bottleneck before the GNN forward pass? Does this imply a fundamental limitation for DistMLIP's performance on future models that might incorporate even higher-order (e.g., four-body) interactions?

**Details Of Ethics Concerns:**

No Ethics Concerns

---

> ### Author Response · Authors · 2025-11-13
>
> Thank you for the rigorous review! We will incorporate your feedback into the manuscript. Here are responses to your weaknesses and questions sections:
>
> >The current work is an excellent demonstration of single-node, multi-GPU scaling. Could you elaborate on the primary challenges for extending DistMLIP to a multi-node environment? Specifically, how do you envision managing the atom_transfer step across a network interconnect, and what do you anticipate will be the new performance bottleneck (e.g., network latency vs. bandwidth)?
>
> The primary challenge for extending DistMLIP to a multi-node environment involves efficient peer to peer communication between GPUs on different nodes. A multi-node implementation would require additional consideration on the specific network topology and how, physically, the nodes are interconnected. This additional consideration is necessary due to the stringent latency requirements for each individual timestep as a result of the integration of millions to billions of such timesteps during simulation. We anticipate the new performance bottleneck to undoubtedly be network latency during inter-GPU feature transfers.
>
> >The abstract and introduction claim "up to 8x faster". However, the direct end-to-end inference comparison with SevenNet shows a ~4x speedup , and the LAMMPS-MACE comparison shows similar (not 8x faster) speeds, albeit with a non-compiled model. The 8x speedup figure appears to be sourced from the partitioning algorithm comparison in Table 4. The authors should clarify this in the main text to avoid overstating the end-to-end simulation speedup.
>
> Thank you for the insight, we will clarify this in the main text.
>
> >The suboptimal weak scaling of CHGNet is due to the three-body graph construction. Is this construction step (described in Algorithm 2) fully parallelized within DistMLIP, or is it a separate, serial (or partially parallel) step that acts as a bottleneck before the GNN forward pass? Does this imply a fundamental limitation for DistMLIP's performance on future models that might incorporate even higher-order (e.g., four-body) interactions?
>
> The three-body graph construction algorithm (Algorithm 2) is, by design, a parallel algorithm. However, the bond graph scales to the 6th power with respect to the number of nodes within the system. As a result, graph construction, partitioning, and information transfer all scale accordingly. We do not believe this to be a fundamental limitation for DistMLIP performance on future models as three-body and higher-order graphs are becoming increasingly rare. Most models rely purely on atom graphs as well as spherical convolutions featured in the MACE, TensorNet, and eSEN models.
>
> >The eSEN model consistently performs poorly, with high memory usage and frequent OOM errors. While this is likely due to the model's architecture rather than DistMLIP, its poor performance slightly detracts from the platform's "general and versatile" claim.
>
> We believe that the future of MLIP applications will require both fast distributed platforms such as ours as well as model architectures that are both accurate and efficient. As a result, we believe the efficiency of eSEN to be independent and out of scope for our current work.
>
> Once again, we would like to thank you for the rigorous review!

---

### Author Response · Authors · 2025-11-14
**Uploaded revised version of the paper**

We would like to thank all reviewers for their reviews!

We're glad the reviewers were impressed with the state of the art performance (in both speed and scale) of DistMLIP as well as its generality and ease of use. We have uploaded a new PDF incorporating all changes in response to the reviewer’s suggestions.


Our modifications, highlighted in blue text in the pdf, are as follows:


From Reviewer dy5L:
* Added clarification in the main text that the 8x speedup refers to the partitioning algorithm comparison in Table 4.

From Reviewer 9NL9:
* Incorporated plots showing the breakdown of inference time between forward pass, backward pass (necessary for force calculations), data transfer, and graph construction overhead to Appendix E. We provide 2 plots: one showing the breakdown where the total number of atoms are fixed and the number of GPUs increases and one showing the inference breakdown where the number of atoms per GPU is fixed and the total number of GPUs increases.
* Added clarification in Appendix J elaborating on how to implement a new MLIP onto the DistMLIP platform.

From Reviewer SsYg:
* Fixed typos.
* Included example of stable simulation over 2 nanosecond simulation on 8 GPUs (Appendix K).
* Added performance metric of large scale training on classical MD labels using DistMLIP (Appendix M).

We believe that DistMLIP will enable scientists and practitioners to study physical phenomena at scales that were unprecedented before – scales that were previously unstudy-able using molecular dynamics simulation at high accuracy – and accelerate scientific discovery ranging from protein dynamics all the way to battery cathode discovery.

---

### Meta-Review · Area_Chair_AdVa · 2026-01-08

**Summary:**

The paper presents DistMLIP, a distributed inference platform designed to scale machine-learning interatomic potentials (MLIPs) to very large atomistic systems, where the parallelization partitioning is on the graph level (vs. spatial).

Reviewers acknowledge the value and timeliness of the contribution to the community, appropriate technical design and implementation, generality over architecture choices, comprehensive evaluation, and impressive efficiency in scaling up system size.

Reviewers also raised a few concerns:
1. The implementation and demonstration is only on single-node architecture, limiting its scalability to larger scale systems.
2. Limited support for MLIP models that involves higher-order interactions.
3. A breakdown analysis on the cost is not sufficiently detailed.
4. Comparison with other distributed models/implementations is not presented.
5. The implementation does not support scaling up training for now.
6. Robustness of the results to the partitioning operation.

**Reviewer Concerns:**

From authors' rebuttal, the concerns are addressed in the following ways:
1. The authors mentioned that a cross-node implementation would involve more than a proper algorithm design and more hardware restrictions. I generally agree with this argument and would perceive a proper design for a single node holds its value. Nevertheless, extending to multiple nodes is always a desire and the authors should consider some simple demonstrations even if the improvement is not as significant.
2. I would perceive an efficient distributed implementation for two-body interaction models already a work with sufficient practical value.
3. In the rebuttal, the authors provided a breakdown cost result with the scaling of the number of GPUs and atoms, which seems to indicate a reasonable support on the overall scaling advantage.
4. I agree that Allegro is a distributed implementation for a specific model architecture. Nevertheless, an accuracy-efficiency could still provide more evidence, e.g., reflecting the arguments on the conceptual comparison with Allegro in Sec. 2.2.
5. I'm not quite convinced about authors' defense that the lack of study for training is simply because lack of training data on large-scale molecules. People are actively scaling up implementation and deriving more scalable algorithms for generating accurate labels on large molecules. Nevertheless, the authors provided a training result in the rebuttal period, and I hope the result could be more clearly described in the main context.
6. The authors provided a stability study in the rebuttal period which seems reasonable.

**Reviewer Scores:**

For the two negatively rating reviewers,
* I would speculate Reviewer 9NL9 to slightly increase the score as additional experimental results are provided consistent with the main claims, which addresses a major concern. There remains concerns but I would perceive it as a minor issue.
* Although Reviewer SsYg leaves a negative score, the sentiment from his/her further replies indicate a relief of major concerns.

As summarized above, there indeed remain some insufficiencies. Nevertheless, I tend to believe the major contributions are already worth a publication.

---

### Decision · Program_Chairs · 2026-01-26

Accept (Poster)